

# A comparative analysis of the rhizosphere microbial communities among three species of the *Salix* genus

Tianqing Feng[1,2,*], Juan Li[3,*], Xiaoning Mao[1,2], Xionglian Jin[1,2], Liang Cheng[3], Huichun Xie[1,2] and Yonggui Ma[1,2]

[1] Key Laboratory of Medicinal Plant and Animal Resources of the Qinghai-Tibetan Plateau in Qinghai Province, Xining, China
[2] School of Life Science, Qinghai Normal University, Xining, China
[3] Qinghai Academy of Agriculture and Forestry Sciences, Qinghai University, Xining, China
* These authors contributed equally to this work.

Corresponding authors
Huichun Xie, yezino.1@163.com
Yonggui Ma, 1261602778@qq.com

## ABSTRACT

Rhizosphere microorganisms exert a significant influence in counteracting diverse external stresses and facilitating plant nutrient uptake. While certain rhizosphere microorganisms associated with *Salix* species have been investigated, numerous rhizosphere microorganisms from various *Salix* species remain underexplored. In this study, we employed high-throughput sequencing to examine the rhizosphere bacterial and fungal communities composition and diversity of three *Salix* species: *Salix zangica* (SZ), *Salix myrtilllacea* (SM), and *Salix cheilophila* (SC). Furthermore, the BugBase and FUNGuild were utilized to predict the functional roles of bacterial and fungal microorganisms. The findings revealed notable variations in the alpha and beta diversities of bacterial and fungal communities among the three *Salix* species exhibited significant differences ($p < 0.05$). The relative abundance of *Flavobacterium* was highest in the SZ samples, while *Microvirga* exhibited significant enrichment in the SM samples. *Microvirga* and *Vishniacozyma* demonstrate the highest number of nodes within their respective bacterial and fungal community network structures. The functions of bacterial microorganisms, including Gram-positive, potentially pathogenic, Gram-negative, and stress-tolerant types, exhibited significant variation among the three *Salix* species ($p < 0.05$). Furthermore, for the function of fungal microbe, the ectomycorrhizal guild had the highest abundance of symbiotic modes. This results demonstrated the critical role of ectomycorrhizal fungi in enhancing nutrient absorption and metabolism during the growth of *Salix* plants. Additionally, this findings also suggested that *S. zangica* plant was better well-suited for cultivation in stressful environments. These findings guide future questions about plant-microbe interactions, greatly enhancing our understanding of microbial communities for the healthy development of *Salix* plants.

## INTRODUCTION

The rhizosphere plays a vital role in supplying water and nutrients to plants (*Lei et al., 2023*), and is also the most active habitat for microorganisms. The rhizosphere microbiome

not only facilitates enhanced nutrient absorption in plants but also modulates their resistance to abiotic stress (*Naylor & Coleman-Derr, 2018*), such as the drought stress (*Liu et al., 2021*; *Fang et al., 2023*), water stress (*Bhattacharyya et al., 2021*; *Agoussar et al., 2021*), salt stress (*Wang et al., 2023*; *Qu et al., 2024*), heavy metal stress (*Wang et al., 2021a*; *Jiao et al., 2023*), cold stress (*Zhang et al., 2024*). The interplay between plants and microorganisms is intricate (*Mendes et al., 2011*; *Busby et al., 2017*). Consequently, comprehending the composition and roles of microbial communities in the plant rhizosphere is essential for promoting plant growth (*Zhang et al., 2021*).

The genus *Salix*, commonly known as willows, encompasses between 330 and 500 species, along with over 200 hybrids (*Isebrands & Richardson, 2014*). This genus is characterized by rapid growth rates, substantial biomass production, and remarkable environmental adaptability, including resistance to salinity and drought as well as the capacity for heavy metal adsorption (*Yang et al., 2021*; *Guo et al., 2022*; *Ran et al., 2021*). The extracts and secondary metabolites derived from willow plants have been extensively researched, exhibiting a range of functions including antioxidant, anti-inflammatory, antiproliferative, and antibacterial properties (*Tawfeek et al., 2021*). Salix microbial communities have been extensively studied in different environments, such as floodplain (*Hashimoto & Higuchi, 2003*), cultivated land (*Hrynkiewicz et al., 2012*), and polluted land (*Dagher, Pitre & Hijri, 2020*).

The capacity of various willow genotypes is intricately linked to the composition of their rhizosphere microbial communities (*Yergeau et al., 2015*, *2018*). Previous studies have documented the microbial community structure of the rhizosphere soil associated with various *Salix* species, including *Salix linearistipularis* (*Cui et al., 2023*), *Salix purpurea* (*Alotaibi et al., 2021*), *Salix psammophila* (*Liang et al., 2021*), *Salix sachalinensis* (*Hashimoto & Higuchi, 2003*), *Salix viminalis*, *Salix dasyclados*, *Salix schwerinii*, *Salix caprea*, *Salix fragilis*, *Salix × mollissima* (*Hrynkiewicz et al., 2012*), as well as *Salix jiangsuensis* and *Salix × aureo-pendula* (*Wang et al., 2021b*). Although the rhizosphere microorganisms associated with these *Salix* species have been investigated, numerous other species of rhizosphere microbe remain unexplored.

*Salix zangica* is a low-growing shrub willow that typically thrives at altitudes of 4,500 m and originates from the eastern region of Xizang, China (*Ma, Ye & Li, 2019*). *Salix myrtillacea* thrives in regions with elevations between 2,700 and 4,800 m. Research indicates that female *S. myrtillacea* exhibit greater drought tolerance compared to their male counterparts (*He et al., 2023*). Furthermore, the impact of exogenous amino acids on physiological responses, rhizosphere microbial composition, and metabolic processes of *S. myrtillacea* under drought stress was investigated (*Kong et al., 2022*). *Salix cheilophila* is a xerophytic species and one of the few plants capable of thriving in saline-alkaline soils (*Yu et al., 2014*). At present, there is a paucity of research on rhizosphere microbial community of *S. zangica*, *S. myrtillacea*, and *S. cheilophila*.

A comprehensive understanding of the rhizosphere microbial communities associated with *Salix* species can provide valuable insights into plant-microbe interactions, significantly enhancing our knowledge of microbial communities essential for the healthy development of *Salix* plants. However, limited information is currently available regarding

the rhizosphere microbial communities of *S. zangica*, *S. myrtillacea*, and *S. cheilophila*. The objective of the current study was to characterize the rhizosphere soil bacterial and fungal communities associated with three species of the *Salix* genus, specifically *S. zangica*, *S. myrtillacea*, and *S. cheilophila*. This research aimed to: (1) compare the diversity of bacterial and fungal communities in the rhizosphere soil across different genotypes of the *Salix* genus; (2) analyze the composition of bacterial and fungal communities among these three *Salix* species; and (3) assess functional differences among various *Salix* species, thereby elucidating how plant genotypes influence rhizosphere soil microorganisms.

# MATERIALS AND METHODS

## Samples collection

The sampling site for the experiment is located at Huangzhong County, Xining City, Qinghai Province, China (36.617047°N, 101.399316°E). This district has a continental climate, with an average annual temperature of 5.1 °C, an annual precipitation of 509.8 mm, an evaporation of 900–1,000 mm, an average frost-free period of 170 days and 2,453 h of sunshine. In May 2023, a total of nine rhizosphere soil samples were collected from *S. zangica* (SZ), *S. myrtillacea* (SM), and *S. cheilophila* (SC), all grown as annual cuttings within the same agricultural field (Fig. S1). At each sampling point, soil samples were randomly collected from a depth of 0 to 15 cm using the five-point sampling method. For each species, three replicate rhizosphere soil samples, each weighing 50 grams, were collected. The distance between sampling points of the same species was maintained at a minimum of 10 m. Sterile gloves and a sterile shovel were used during the sampling process. The collected samples were placed in sterile self-sealing bags, transported under low-temperature conditions, and subsequently frozen at −80 °C upon arrival at the laboratory.

## Soil DNA extraction and PCR amplification

The genomic DNA from the microbial community was extracted from rhizosphere soil samples utilizing the E.Z.N.A. Soil DNA Kit (Omega Bio-tek, Norcross, GA, USA) in accordance with the manufacturer's protocols. DNA extracts were analyzed using a 1% agarose gel, and the concentration and purity of the DNA were assessed with a NanoDrop 2000 UV-Vis spectrophotometer (Thermo Scientific, Wilmington, DE, USA). The V3-V4 region of bacterial 16S rRNA was amplified utilizing primers 338 (5′-ACTCCTACGG AGGCAGCAGCAG-3′) and 806R (5′-GGACTACHVGGGTWTCTAAT-3′) (*Klindworth et al., 2013*). The fungal ITS region was amplified with primers ITS1F (5′-CTTGG TCATTTAGAGGAAGTAA-3′) and ITS2R (5′-GCTGCGTTCTTCATCGATGC-3′) (*Yan et al., 2021*). The PCR amplification conditions were as follows: an initial denaturation at 95 °C for 3 min, followed by a denaturation step at 95 °C for 30 s, annealing at 55 °C for 30 s, extension at 72 °C for 45 s, a final extension phase at 72 °C for 10 min, and storage at a temperature of 4 °C (*Li et al., 2024*). The PCR products of 16S and ITS were dispatched to Megi Biological Technology Co., Ltd. in Shanghai, China for high-throughput sequencing.

## Bioinformatic and statistical analysis

The paired-end reads generated through Illumina sequencing were concatenated using Flash (version 1.2.11). Quality control and filtering of the reads were performed based on sequencing quality metrics. The high-quality sequences were clustered into operational taxonomic units (OTUs) at a 97% similarity threshold using USEARCH v11 (*Edgar, 2010*), from which representative sequences for each OTU were selected for further annotation. Taxonomic annotations for bacteria and fungi were conducted using the Silva database (version 138) and the ITS database (version UNITE 8.0), respectively (*Koljalg et al., 2013*).

Venn diagrams were employed to analyze the shared and unique OTU counts among different groups of samples. The alpha diversity indices, including the Shannon index, Chao index, and Simpson index, were utilized to assess the diversity of bacterial and fungal communities. Principal coordinates analysis (PCoA) was conducted to evaluate the similarity of bacterial and fungal communities from different groups of samples based on Bray-Curtis distance at the OTU level (*Masocha et al., 2022*). The significant differences in microbial community structure among different sample groups were assessed using the Analysis of Similarity (ANOSIM). Additionally, linear discriminant effect size (LEfSe) analysis was employed to identify significantly enriched species among different groups of samples at the genus level (*Segata et al., 2011*). Network analysis was employed to elucidate the interaction relationships among various sample groups. The functional differences in bacterial and fungal communities across different sample groups were predicted using BugBase and FUNGuild, respectively (*Chen et al., 2020*). Box plots for the Shannon, Chao, and Simpson indices were generated utilizing Origin 2018. All data analyses were performed using IBM SPSS Statistics, version 25.0 (IBM Corp., Armonk, NY, USA).

# RESULTS

## The analysis of OTU units

The number of bacterial OTUs in different species rhizosphere of *Salix* genus was much greater than that of fungal taxa (Fig. 1). The number of shared OTUs in the bacterial and the fungal communities was 3,411 (33.02%) and 323 (22.48%), respectively, among the three different species of the *Salix* genus. The number of unique OTUs in the bacterial community was 1,382, 1,631, and 1,664 in the SZ, SM, and SC samples, respectively, accounting for 13.38%, 15.79%, and 16.11% of all OTUs (Fig. 1A). The number of unique OTUs in the fungal community was 324, 292, and 158 in the SZ, SM, and SC samples, respectively, accounting for 22.55%, 20.32%, and 11.00% of the total OTUs (Fig. 1B).

## The analysis of alpha and beta diversities

To assess the alpha diversity, the Chao index, Shannon index, and Simpson index were computed and presented in Fig. 2. The Chao index and Shannon index of the bacterial community were higher than those of the fungal community, whereas the Simpson index of the bacterial community was lower than that of the fungal community (Fig. 2). Significant differences were detected in the Simpson index and Shannon index of the bacterial and fungal communities among the three species samples ($p < 0.05$), but no

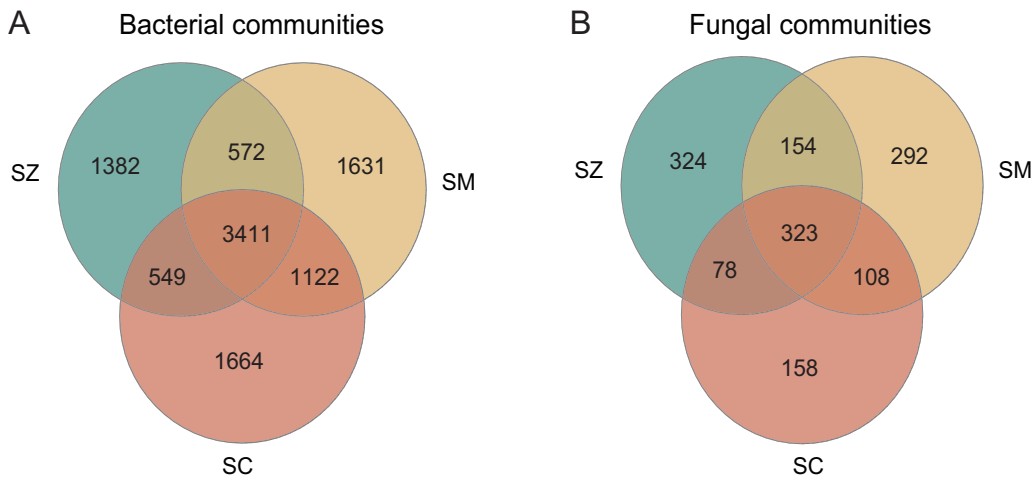

**Figure 1** The analysis of OTUs in the bacterial communities (A) and the fungal communities (B).

**Figure 2** The α-diversity analysis of the bacterial community (A) and the fungal community (B). Three α-diversity index including Chao, Simpson, and Shannon ind ices were calculated. The Kruskal-Wallis H test was employed to assess significant differences among SZ, SM, and SC groups. The SZ, SM, and SC represent *Salix zangica*, *Salix myrtilllacea*, and *Salix cheilophila* samples, respectively.

significant differences were identified in the Chao index of the bacterial and fungal communities among the three species samples ($p > 0.05$).

The PCoA of the bacterial community exhibited that Axis1 explained 52.60% and Axis2 explained 17.88% of the variation, respectively (Fig. 3A). The PCoA of the fungal community exhibited that Axis1 explained 54.14% and Axis2 explained 32.08%% of the variation, respectively (Fig. 3B). The result indicated that whether in the bacterial community or the fungal community, the different types of *Salix* genus were clearly and distinctly clustered and presented a significant difference ($p < 0.05$) (Fig. 3).

## The composition of bacterial and fungal communities

A total of 12,680 OTUs were detected in all bacterial community samples and were assigned to 41 phyla, 137 classes, 361 orders, 580 families, 1,147 genera, and 2,722 species. The dominant bacterial phyla (relative abundance > 1%) were Actinobacteriota, Proteobacteria, Acidobacteriota, Chloroflexi, Firmicutes, Bacteroidota, Myxococcota, Gemmatimonadota, Verrucomicrobiota, Planctomycetota, and Cyanobacteria in samples of three species of the *Salix* genus (Fig. 4A). The relative abundance of Actinobacteriota (32.2%) in the SZ samples was higher than that in the SM (20.0%) and SC (24.3%) samples.

The predominant genera (relative abundance > 1%) were *Pseudarthrobacter*, *Bacillus*, *Cryobacterium*, *Sphingomonas*, *Gaiella*, *Flavobacterium*, *Nocardioides*, *Sporosarcina*, *Skermanella*, RB41, *Microvirga*, *Paenisporosarcina*, *Streptomyces*, *Ilumatobacter*, and *Ornithinibacter* (Fig. 4B). The relative abundances of *Cryobacterium*, *Flavobacterium*, and *Ornithinibacter* were higher in the SZ (3.6%, 3.0%, and 1.3%) samples than those in the SM (0.6%, 0.3%, and 0.3%) samples and the SC (0.7%, 0.7%, and 0.3%) samples, respectively. The relative abundance of *Streptomyces* was lower in the SM (0.4%) and SZ (0.4%) samples than that in the SC (1.5%) samples.

A total of 2,945 OTUs were identified in the fungal community samples, classified into 16 phyla, 48 classes, 111 orders, 252 families, 576 genera, and 977 species. The predominant fungal phyla (relative abundance > 1%) were Ascomycota, Basidiomycota, and Mortierellomycota (Fig. 5A). Ascomycota was the most abundantly dominant phylum in all samples, with the relative abundances being 79.3%, 66.4%, and 89.4% in the SZ, SM, and SC samples, respectively. The top five fungal genera in terms of relative abundance were *Peziza*, *Thelebolus*, *Mortierella*, *Geopora*, and *Didymella* (Fig. 5B). The relative abundances of *Peziza* and *Acaulium* were higher in the SC samples than those in the SZ and SM samples. The relative abundances of *Thelebolus*, *Phoma*, *Tausonia*, *Vishniacozyma*, *Coprinellus*, *Naganishia*, *Chaetasbolisia*, and *Lophiotrema* were greater in the SZ samples than those in the SC and SM samples. The relative abundances of *Pulvinula*, *Tomentella*, *Inocybe*, *Sepultariella*, and *Hebeloma* were higher in the SM samples than those in the SZ and SC samples.

## The species difference analysis of bacterial and fungal communities

For the bacterial community, when the linear discriminant analysis (LDA) > 3, there are 11, five, and two significant difference genera in the SZ, SM, and SC samples, respectively (Fig. 6A). The relative abundance of *Flavobacterium*, *Nocardioides*, and *llumatobacter* were

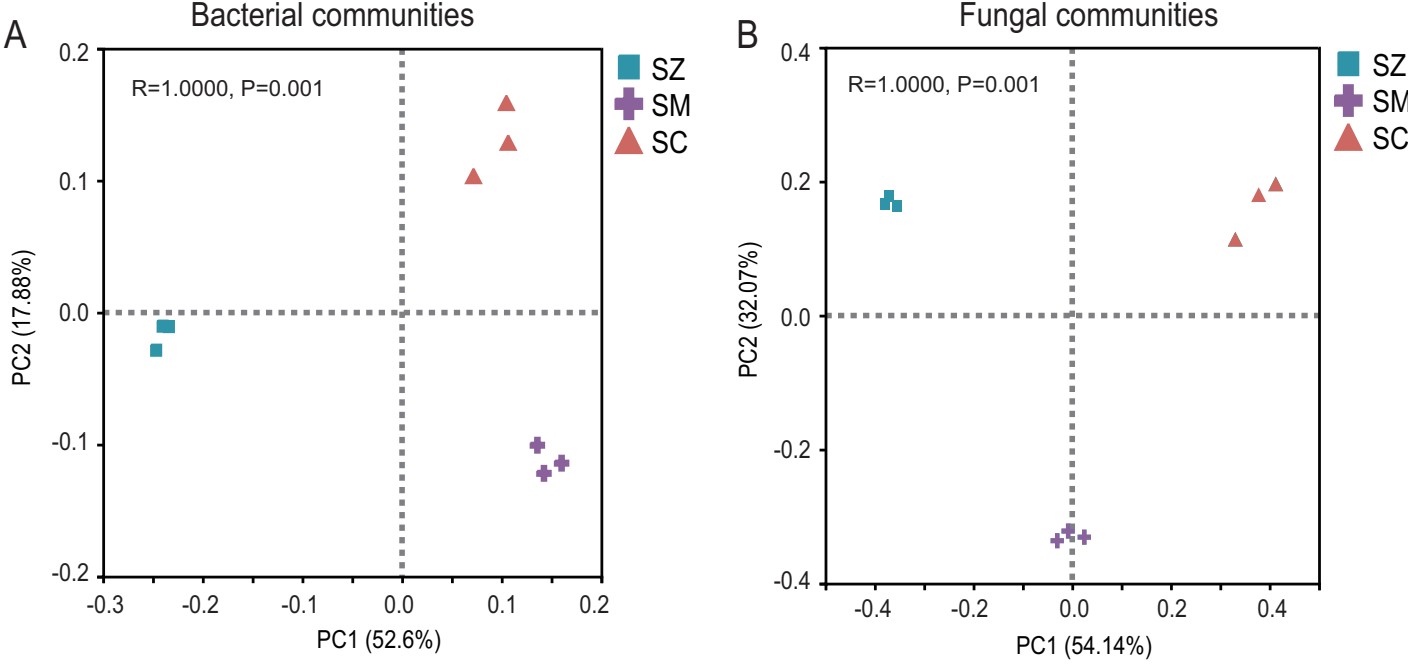

**Figure 3 Principal coordinate analysis (PCoA) based on the Bray-Curtis index show ed the differences in bacterial (A) and fungal (B) communities among SZ, SM, and SC groups.** The SZ, SM, and SC represent *Salix zangica*, *Salix myrtilllacea*, and *Salix cheilophila* samples, respectively.

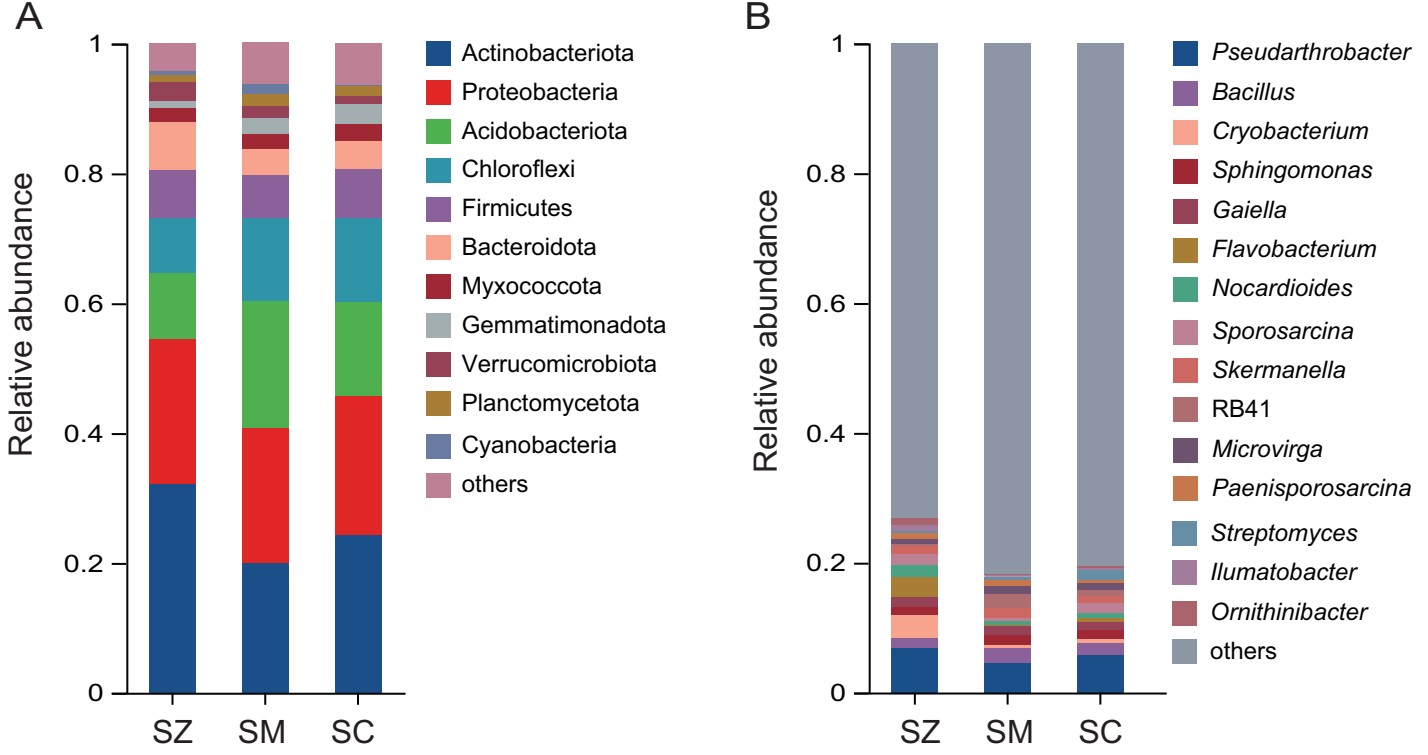

**Figure 4 The relative abundance of the bacterial community in three distinct species of the *Salix* genera samples.** (A) At the phyla level; (B) at the genus level.

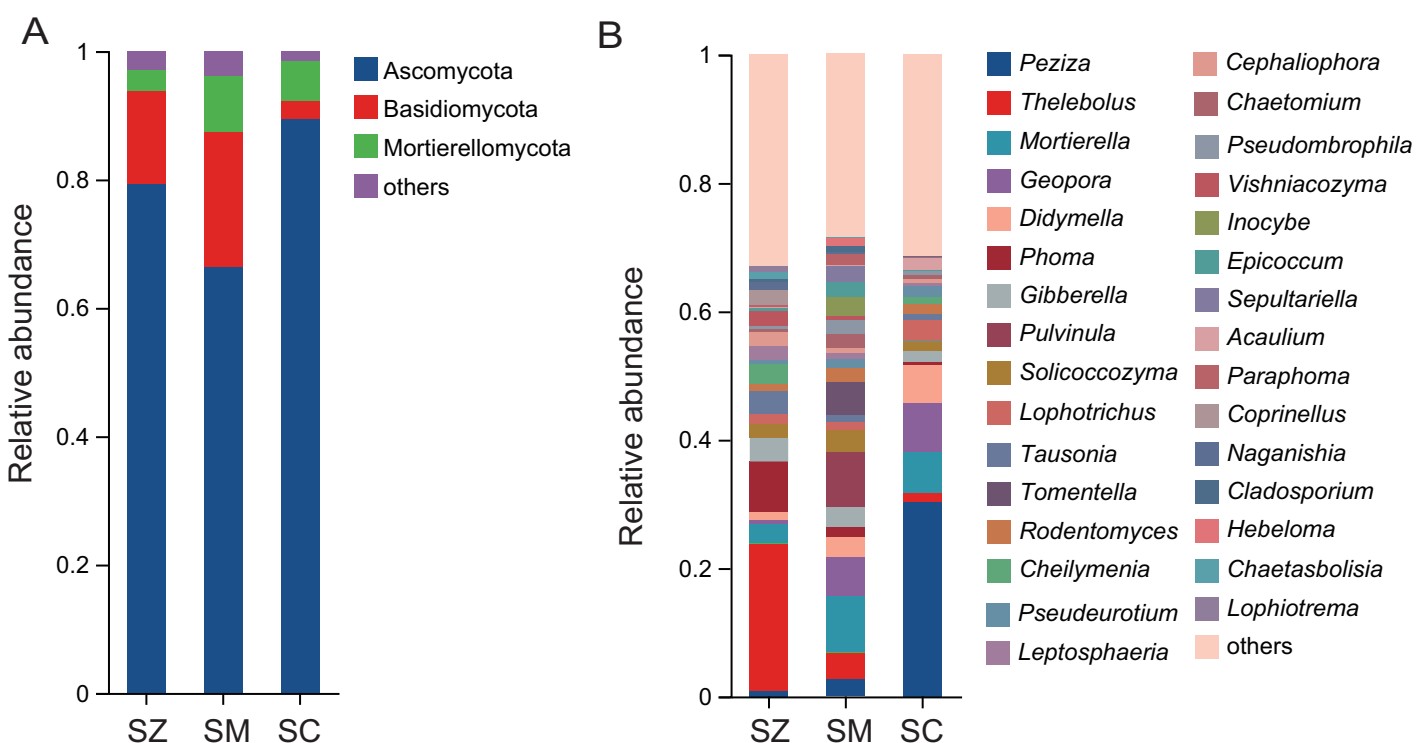

**Figure 5  The relative abundance of the fungal community in three different species of the *Salix* genera samples.** (A) At the phyla level; (B) at the genus level.

higher in the SZ samples, *Microvirga* was significantly enriched in the SM samples, and *Mycobacterium* was significantly enriched in the SC samples (Fig. 6A).

For the fungal community, when the LDA > 3, there are 22, 23, and eight significantly difference genera in the SZ, SM, and SC samples, respectively (Fig. S2). The *Thelebolus*, *Phoma*, *Hapsidospora*, and *Vishniacozyma* were significantly enriched in the SZ samples, the *Pulvinula*, *Mortierella*, and *Epicoccum* were significantly enriched in the SM samples, and the *Peziza* were significantly enriched in the SC samples (LDA > 4) (Fig. 6B).

## The network analysis of bacterial and fungal communities

The top 50 taxa were used to construct the network at the genus level. Compared with the fungal network structure, the bacterial community network structure has more significantly related species, and is also more complex and stable (Fig. 7). For the bacterial community, there are 186 positive correlations and 177 negative correlations were identified from 50 nodes, which belonged to Chloroflexi, Acidobacteriota, Actinobacteriota, Proteobacteria, Planctomycetota, Bacteroidota, Gemmatimonadota, Firmicutes, Verrucomicrobiota, and Nitrospirota. The degrees of *Microvirga*, *Nocardioides*, *Bryobacter*, and *Marmoricola* nodes were 26, 25, 25, and 25, respectively, which showed the higher number than other nodes. For the fungal community, there are 215 positive correlations and 89 negative correlations were found, which were distributed among Ascomycota, Basidiomycota, and Mortierellomycota. The *Vishniacozyma* has the highest number (21) than other nodes.

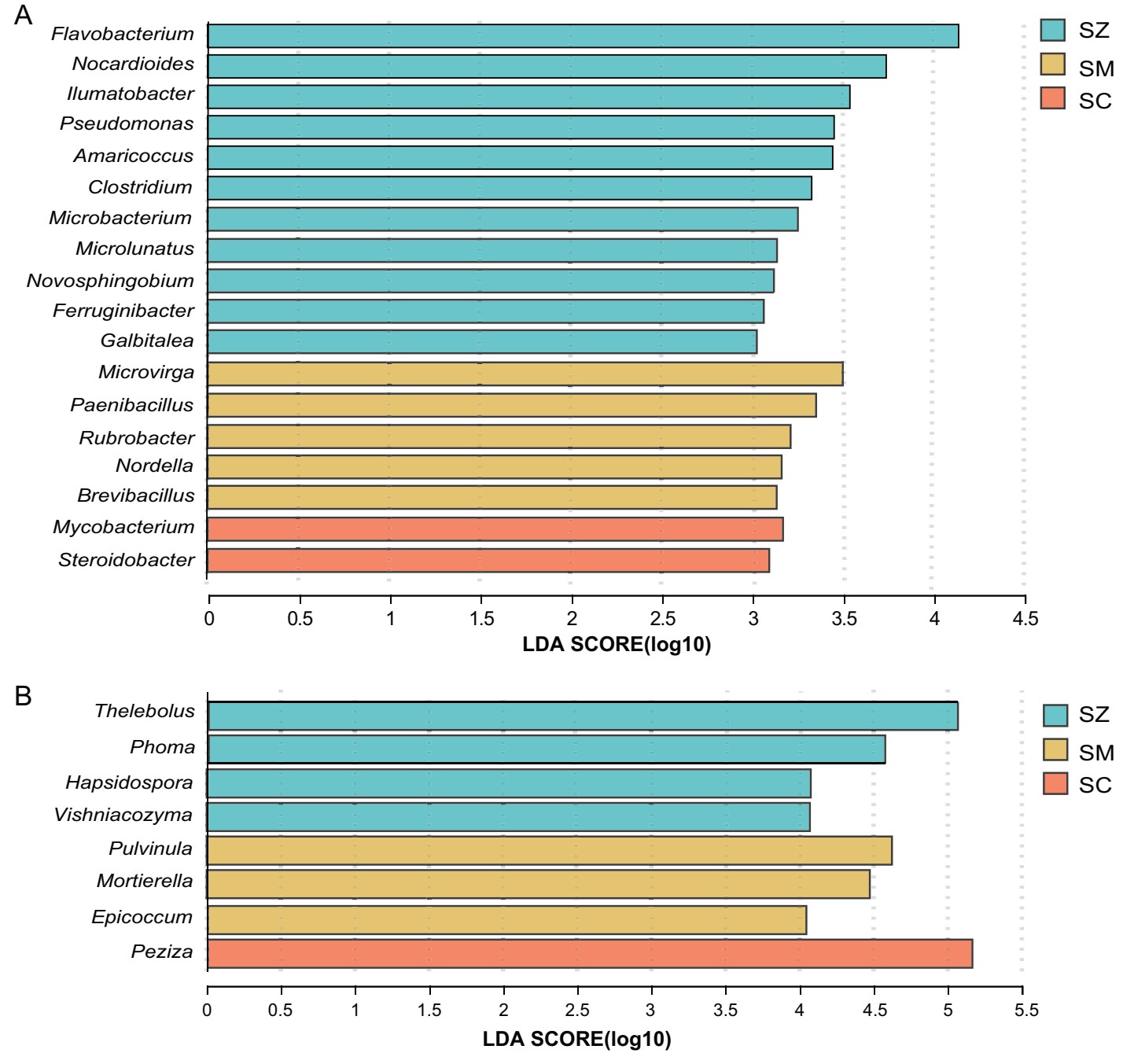

**Figure 6** Linear discriminant analysis effect size (LEfSe) results revealed the significantly different microorganisms at the genus level for (A) bacterial communities (LDA > 3) and (B) fungal communities (LDA > 4) among SZ, SM, and SC groups. The SZ, SM, and SC represent *Salix zangica*, *Salix myrtilllacea*, and *Salix cheilophila* samples, respectively.

## The function prediction of bacterial and fungal communities

The bacterial communities phenotypes of three species of the *Salix* genus were predicted using BugBase, including Gram Positive, Gram Negative, Biofilm Forming, Potentially Pathogenic, Mobile Element Containing, Oxygen utilizing (Aerobic, Anaerobic, Facultatively Anaerobic), Oxidative Stress Tolerant (Fig. 8A). Gram Positive, Potentially

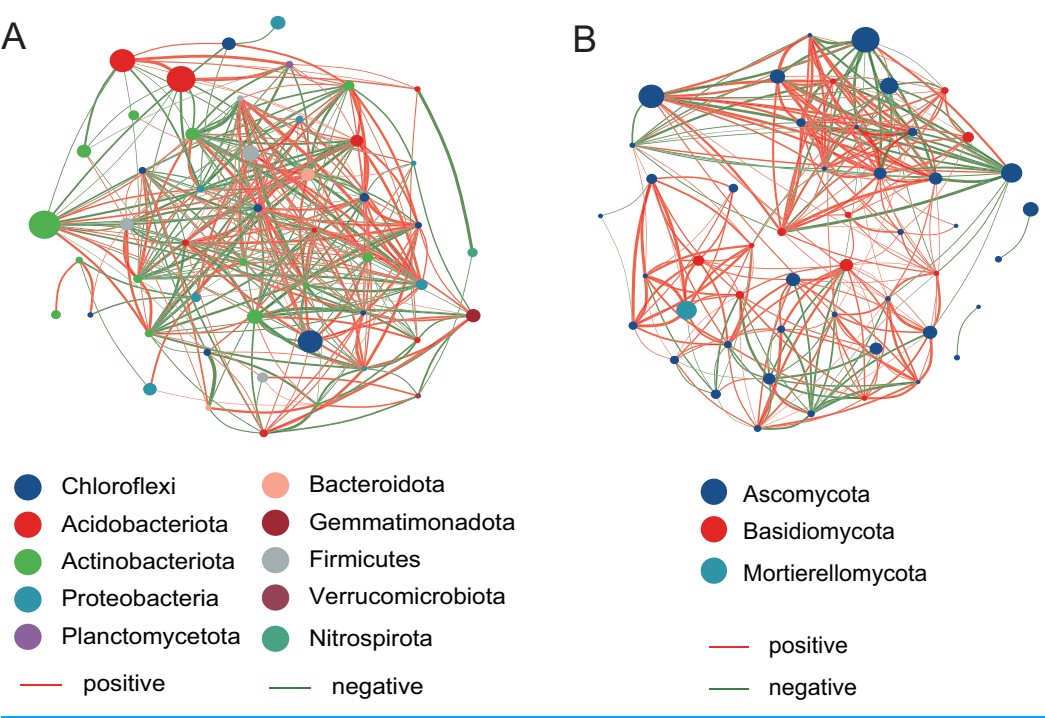

**Figure 7 The network analysis of the bacterial community (A) and the fungal community (B).**

Pathogenic, Gram Negative, and Stress Tolerant were found to exhibit a significant difference among the three species of the *Salix* genus ($p < 0.05$). For the Gram Negative and Potentially Pathogenic phenotypes, the relative abundance of RB41 was significant higher in the SM samples than that in the SC and SZ samples, and the relative abundance of *Nitrospira* was greater in the SC samples than that in other samples (Figs. S3A and S3C). Besides, for the Gram Positive and Stress Tolerant phenotypes, the relative abundance of *Cryobacterium* was obvious greater in the SZ samples than other samples (Figs. S3B and S3D).

FUNGuild was employed to predict the function of the fungal community of three *Salix* genus. The results showed that nine trophic mode groups could be classified, with Saprotroph-Symbiotroph, Saprotroph, Symbiotroph, Pathotroph-Saprotroph-Symbiotroph, Pathotroph, Pathotroph-Saprotroph, Pathogen-Saprotroph-Symbiotroph, Pathotroph-Symbiotroph, and Saprotroph-Pathotroph-Symbiotroph (Fig. 8B). The Saprotroph-Symbiotroph, Saprotroph, Symbiotroph, and Pathotroph-Saprotroph-Symbiotroph were the major components, with the percentages of OTUs in these major functional categories being 15.75–38.52%, 14.18–27.25%, 0.71–24.32%, and 7.22–22.64%, respectively. The Ectomycorrhizal Guild exhibited the highest abundance of symbiotic modes across all samples (Table S1).

# DISCUSSION

To enhance our understanding of the variations in rhizosphere microorganisms of *Salix* genus, this study investigated the diversity, structure, and functional characteristics of

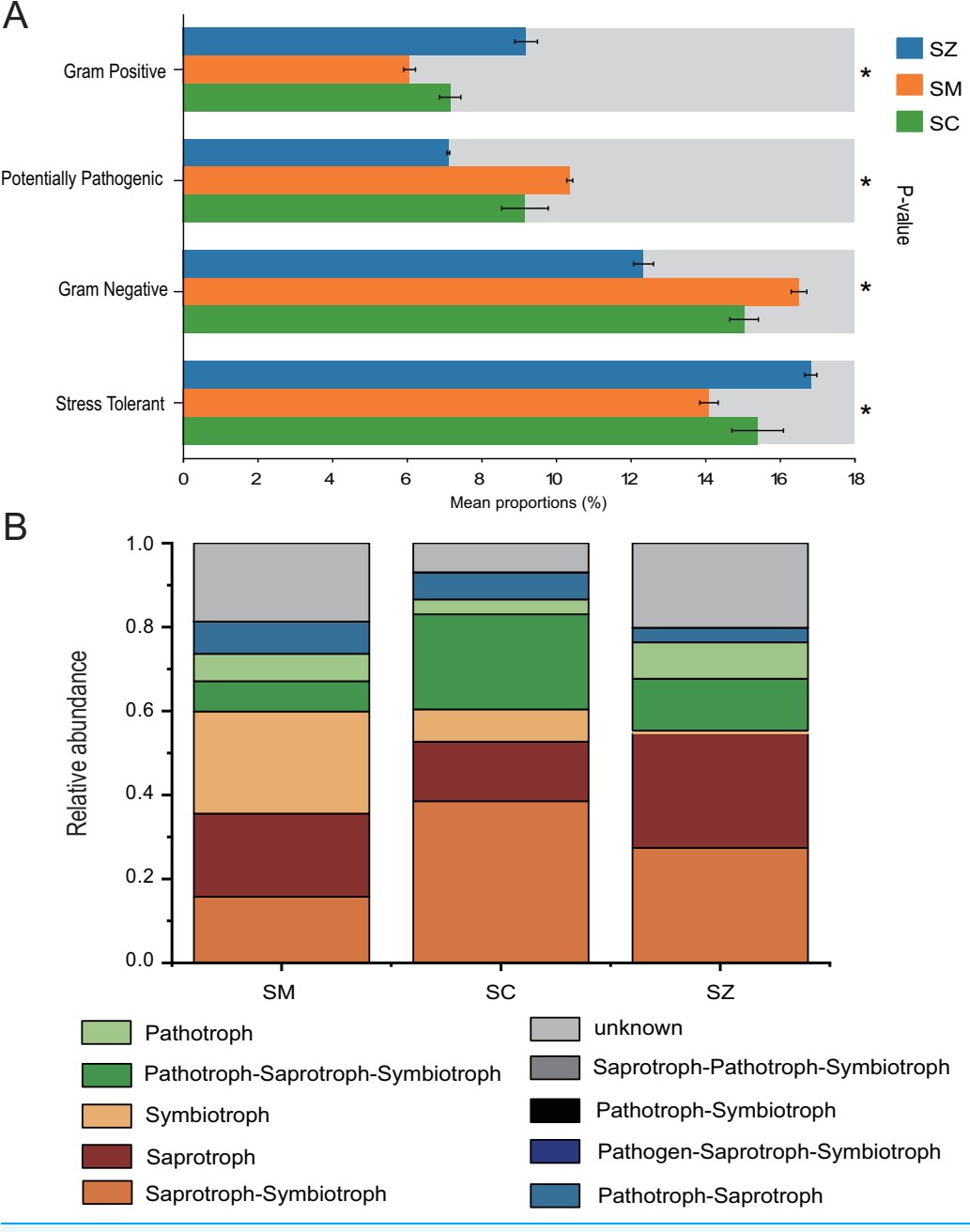

**Figure 8 The BugBase and FUNGuild were employed to predict the function of bacterial community (A) and fungal community (B) among SZ, SM, and SC groups, respectively.** SZ, SM, and SC represent *Salix zangica*, *Salix myrtilllacea*, and *Salix cheilophila* samples, respectively. An asterisk (*) indicates that the *p*-value is less than 0.05.

microbial communities of three species of *Salix* genus. Microbial communities are essential for the maintenance of soil ecosystems and serve as sensitive biomarkers for evaluating soil health and functionality (*Zuppinger-Dingley et al., 2014*).

This study demonstrated that the alpha diversity and beta diversity of bacteria and fungi exhibited significant differences among the three species of the *Salix* genus. This is
consistent with the previously reported results of rhizosphere microbial communities of blueberry varieties (*Zhang et al., 2021*). The diversity of bacterial and fungal communities was observed to be greatest in the SM samples, indicating that these samples harbor a more extensive array of microbial communities. In this study, three *Salix* species were cultivated in the same field under identical environmental conditions to eliminate potential external influences. It has been established that the rhizosphere microbial community associated with *Salix* plants is correlated with their phylogeny (*Bell et al., 2014*). Therefore, our results also indicated that the host genotype exerts significant effects on the microbial communities associated with *Salix* plants, where the roots exude various substances and attract different types of microbial communities (*Jacoby et al., 2017*). In addition, the PCoA results displayed that the genetic variation among species within the *Salix* genus explains 70.48% of the variability in bacterial communities and 86.22% of the variability in fungal communities. In comparison to rhizosphere bacterial communities, rhizosphere fungal communities exhibit a greater influence from plant genotypes, likely attributable to the more intimate relationship between fungal communities and their host plants (*Deng et al., 2018*). Previous study also reported that soil fungal communities are significantly shaped by the genotypes or species of plants (*Prescott & Grayston, 2013*).

In our study, the bacterial phyla with the highest relative abundance in *S. zangica*, *S. myrtillacea*, and *S. cheilophila* were Actinobacteriota, followed by Proteobacteria and Acidobacteriota. This finding aligns with the report on the rhizosphere bacterial community of *S. linearistipularis* (*Cui et al., 2023*); however, it contrasts with the rhizosphere bacterial community of *Tamarix chinensis*, where Proteobacteria was predominant, followed by Actinobacteria and Firmicutes (*Qu et al., 2024*). This discovery further substantiates that, when contrast with plants from distinct genera, plants within the same genus are more inclined to recruit rhizosphere microbiota that are more similar in composition. In addition, the Ascomycota was the most abundantly dominant fungal phylum in all samples, with proportions of 79.3%, 66.4%, and 89.4% in SZ, SM, and SC samples, respectively. Ascomycota fungi are crucial drivers of carbon and nitrogen cycling in grasslands and shrublands ecosystems, these fungi play significant roles in soil stabilization, plant biomass decomposition, and endophytic interactions with plants (*Challacombe et al., 2019*).

In the genus level, our research findings revealed that *Pseudarthrobacter* exhibited the highest relative abundance across all samples. This significant prevalence can likely be attributed to the genus's capabilities in nitrogen fixation, phosphate and potassium solubilization, as well as the synthesis of indole-3-acetic acid, all of which contribute to promoting plant growth and playing a vital role in the development of *Salix* species (*Ham et al., 2022*; *Issifu et al., 2022*). In addition, the relative abundance of *Cryobacterium*, and *Flavobacterium* was found to be higher in the SZ samples compared to other samples. Previous studies have indicated that various *Cryobacterium* species flourish in low-temperature environments (*Niu et al., 2023*; *Liu et al., 2019*, *2020*; *Liu, Yang & Xin*, *2023*). The genome of *Cryobacterium* encompasses an extensive array of stress response genes associated with environmental adaptation, these genes potentially encode specialized proteins or enzymes that assist in preserving cellular structure and function stability under

low-temperature conditions (*Teoh et al., 2024*). Furthermore, *Flavobacteria* emerged as the predominant genus during the cooling phase of static composting (*Shi et al., 2022*), characterized as a strictly aerobic bacterium with significant cellulose degradation capabilities (*Vikas et al., 2018*; *Di Maiuta et al., 2013*). Other researchers also reported that *Flavobacterium* emerged as the predominant organism in a sulfur-dominated supraglacial spring system (*Trivedi et al., 2018*). The findings of this study align with those of prior research, which indicate that *Cryobacterium* and *Flavobacterium* are the predominant genera in low-temperature environments.

In the fungal community, the relative abundance of *Thelebolus* was higher in the SZ samples compared to other samples. *Thelebolus* exhibits remarkable cold resistance, enabling it to thrive and reproduce in low-temperature environments. In the Antarctic region, *Thelebolus microsporus* was a prevalent fungal species (*Hoog et al., 2005*). Besides, reports indicate that species of *Thelebolus* possess the capability to resist Gram-negative bacteria, specifically *Escherichia coli* (*Ordóñez-Enireb et al., 2022*). *Phoma* was another genus significantly enriched in the SZ samples, and it has been reported that *Phoma* spp. possess antimicrobial properties due to their production of secondary metabolites with antibacterial activity (*Rai et al., 2022*). *Peziza* was significantly enriched in the SC samples, which belongs to the ectomycorrhizal taxa (*Jacquemyn et al., 2016*). Ectomycorrhizal (ECM) fungi establish a symbiotic association with plant roots, significantly enhancing the health and growth of their host plants (*Charya & Garg, 2019*).

To investigate potential interactions between microbial taxa, network analysis of significant co-occurrence patterns among taxa can elucidate the structure of complex microbial communities (*Barberán et al., 2012*). High network complexity serves as an indicator of stable communities (*Mougi & Kondoh, 2012*). Our results demonstrated that the network of bacterial communities exhibited a greater number of nodes and connections compared to fungal communities, suggesting that bacterial community structures were more complex and stable. In the present study, the *Microvirga* genus exhibits the highest number of nodes within the bacterial structural network. *Microvirga*, a symbiotic nitrogen-fixing bacterium, was capable of establishing synergistic relationships with other microorganisms in the soil environment (*Mehmood et al., 2022*; *Msaddak et al., 2019*), which enhances the availability of nitrogen sources for plants, thereby promoting their growth. This may explain the prominence of *Microvirga* in terms of node abundance within the bacterial network structure. The genera *Nocardioides*, *Bryobacter*, and *Marmoricola* represent the second most prominent nodal genera within bacterial network structures. *Nocardioides* are recognized as specialists in the degradation of recalcitrant substances within environmental contexts, demonstrating an ability to thrive in diverse low-nutrient environments (*Ma et al., 2023*). A strain of *Nocardioides* sp. WS12 was isolated from the soil of *Salix alba* and has demonstrated the capability to degrade isoprene (*Larke-Mejia et al., 2019*), the genome contains a comprehensive isoprene monooxygenase gene cluster along with associated genes involved in the isoprene degradation pathway (*Gibson, Larke-Mejía & Murrell, 2020*). Furthermore, the genera *Nocardioids* and *Bryobacter* are significant plant growth-promoting bacteria in the rhizosphere, closely associated with enhancing plant development (*Li et al., 2022*). The *Vishniacozyma*

represents the genus with the highest number of nodes within the fungal network structure. Reports indicated that the *Vishniacozyma Victoriae* strain possesses the capability to solubilize phosphates and inhibit the growth of pathogenic bacteria (*da Silva et al., 2022*; *Sepúlveda et al., 2022*). Therefore, our results indicated that key genera such as *Microvirga*, *Nocardioides*, and *Vishniacozyma* play a crucial regulatory role in maintaining the structure, function, and stability of the soil rhizosphere microbial communities associated with *Salix* plants.

BugBase predicts the functional profiles of bacterial communities associated with three *Salix* species, highlighting significant disparities in the prevalence of Gram-positive, Potentially pathogenic, Gram-negative, and Stress Tolerant bacteria among these *Salix* species ($p < 0.05$). The relative abundance of Stress Tolerance function was significantly higher in the SZ samples compared to the SC and SM samples. The *Cryobacterium* genus was the predominant bacterial group associated with Stress Tolerant phenotypes in the SZ samples (Fig. S3). Furthermore, *Cryobacterium* has been recognized as a dominant genus in cold environments (*Teoh et al., 2024*). Consequently, our findings provide additional evidence that *Cryobacterium* plays a significant role in the rhizosphere soil of *S. zangica*, which is found in mountainous regions at elevations of 4,500 m (*Ma, Ye & Li, 2019*). Besides, the relative abundances of potentially pathogenic and Gram-negative functions were markedly elevated in the SM samples when compared to those observed in the SZ and SC samples. The *Microvirga* genus was significantly enriched in the SM samples, and it has been reported that this genus contains potentially pathogenic species (*Noguera et al., 2018*). However, other studies have also reported that the *Microvirga* genus belongs to rhizobium, which is advantageous for plant growth (*Wan et al., 2024*).

FUNGuild was utilized to predict the functional roles of the fungal community associated with three *Salix* species. The relative abundances of Saprotroph-Symbiotroph and Pathotroph-Saprotroph-Symbiotroph were significantly higher in the SC samples compared to those in SZ and SM samples. A previous study reported that the sites and *Salix* genotype significantly influenced the formation of Ectomycorrhizal fungi (*Hrynkiewicz et al., 2012*). Ectomycorrhizal fungi serve as a protective barrier for plant roots, safeguarding them against pathogenic microorganisms (*Chen et al., 2019*; *Song & Zhou, 2021*). These fungi also were demonstrated enhanced adaptability to extreme environments, including arid, frigid, and desolate conditions (*Allsup, George & Lankau, 2023*; *Zheng & Song, 2022*; *Wu et al., 2023*). In the current study, the Ectomycorrhizal Guild emerged as the predominant symbiotic association across all samples. The abundance of ectomycorrhizal fungi exhibited significant variation among different *Salix* genotypes, with the highest abundance observed in the SM samples.

## CONCLUSIONS

Our findings accentuated the diversity, composition, and functionality of the *Salix* rhizosphere microbiome and illustrated the disparities within the rhizosphere microbiome among different *Salix* species. This study identified key taxa within the bacterial and fungal communities through co-occurrence network analysis, offering a comprehensive perspective on the microbial community in the rhizosphere of *Salix*. The

*Pseudarthrobacter* constituted the predominant genera within the bacterial community in all samples. Moreover, at the genus level, it was observed that there were a greater number of unidentified bacterial taxa across various *Salix* species when compared to the fungal community. This presents an opportunity for further exploration in future research. The investigation of symbiotic microorganisms, particularly ectomycorrhizal fungi, in the interactions between *Salix* plants and microbe in further exploration. Overall, this research contributes to a deeper understanding of the correlation between rhizosphere microbiota and *Salix* plants, and provides evidence for the critical role of ectomycorrhizal fungi in enhancing nutrient absorption and metabolism during the growth of *Salix* plants.

### Funding

This research was funded by the Science and Technology Program of Qinghai Province (grant number 2023-ZJ-908M). The funders had no role in study design, data collection and analysis, decision to publish, or preparation of the manuscript.

### Grant Disclosures

The following grant information was disclosed by the authors:
Science and Technology Program of Qinghai Province: 2023-ZJ-908M.

### Competing Interests

The authors declare that they have no competing interests.

### Author Contributions

- Tianqing Feng performed the experiments, analyzed the data, prepared figures and/or tables, and approved the final draft.
- Juan Li performed the experiments, analyzed the data, prepared figures and/or tables, and approved the final draft.
- Xiaoning Mao analyzed the data, prepared figures and/or tables, and approved the final draft.
- Xionglian Jin analyzed the data, prepared figures and/or tables, and approved the final draft.
- Liang Cheng conceived and designed the experiments, authored or reviewed drafts of the article, and approved the final draft.
- Huichun Xie conceived and designed the experiments, authored or reviewed drafts of the article, and approved the final draft.
- Yonggui Ma conceived and designed the experiments, authored or reviewed drafts of the article, and approved the final draft.

### Data Availability

The original sequences are available at NCBI Sequence Read Archive (SRA): PRJNA1151252.

## Supplemental Information

Supplemental information for this article can be found online at http://dx.doi.org/10.7717/peerj.19182#supplemental-information.

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
