# Peer review of "A comparative analysis of the rhizosphere microbial communities among three species of the Salix genus"

_PeerJ, doi:10.7717/peerj.19182_

## Round 0.1 · original submission · Major Revisions

Please respond to the reviewers' comments and concerns and submit your revised manuscript, together with a point-by-point response letter.

Reviewer 1 ·

Basic reporting

The manuscript meets the criteria to publish in PeerJ

Experimental design

The well-designed experiment is showed in this manuscript

Validity of the findings

1. The abstract is well composed and structured with adequate components. However, in my opinion, the abbreviations of the three Salix species should be changed to be more memorable, such as SZ, SM, and SC for “Salix zangica, Salix myrtilllacea, and Salix cheilophila”, respectively.
2. The introduction is well documented with essential details. However, before the objective statement at the last paragraph, the novelty and necessity of the study should be restated and emphasized.
3. Materials and methods should go along with literatures. Please add citations for each methods and techniques.
4. The sampling method is unclear. Please clarify how the soil was collected, e.g. amount, positions, etc.
5. Figures should be self-explainable. Thus, some figures need notes to explain their components and should not contain abbreviations, such as Figures 2, 3, 6, and 8.

Additional comments

-

·

Basic reporting

This paper explores the rhizosphere bacterial and fungal communities of three Salix species through high - throughput sequencing. It presents innovative and valuable research, with a sound experimental design and comprehensive result analysis.The study focuses on the rhizosphere microbial communities of Salix species, addressing the under - explored nature of most species. This topic is significant and has the potential to contribute to a deeper understanding of plant - microbe interactions. I recommend accepting this manuscript.

Experimental design

The use of high - throughput sequencing in combination with functional prediction tools (BugBase and FUNGuild) is appropriate. These methods can effectively obtain information on microbial community composition, diversity, and functions.

Validity of the findings

The paper clearly shows the differences in the diversity of rhizosphere bacterial and fungal communities, dominant flora, and functional differences among the three Salix species. The results are statistically significant, providing important data for future research.

Reviewer 3 ·

Basic reporting

Dear Editor-in-Chief of the PeerJ / Authors
I have reviewed the manuscript titled “A Comparative Analysis of the Rhizosphere Microbial Communities among Three Species of the Salix Genus.” The authors have effectively addressed the characterization of rhizosphere soil bacterial and fungal communities associated with three Salix species: S. zangica, S. myrtillacea, and S. cheilophila. The manuscript offers a thorough perspective on the microbial communities within the rhizosphere of Salix species, presenting well-defined objectives and contributing valuable insights into the association patterns between rhizosphere soil microorganisms and plant genotypes. It also explores how these associations may vary spatially across different Salix genotypes and how plant genotypes influence rhizosphere microbial communities.

The study is particularly commendable for the significance of its dataset, the robustness of its analyses, and the relevance of its findings. However, the discussion section could be further enriched to align with the scientific standards required by PeerJ for publication.

Additionally, the English language in the manuscript is of high quality, reflecting a strong level of proficiency.
Title: is clear and informative, however, to capture the attention of the reader
Abstract:
Generally, the abstract aligns well with the text and experimental design, and the results are well-presented, even addressing future questions. However, it would be helpful if the authors could elaborate on the implications of their findings, particularly regarding any practical or applicable aspects derived from the results.

Introduction:
Overall, the introduction is well-developed and situates well the readers in the context of the study.
Materials and methods:
This part is present clearly the study setting and taking measurements. The statistical analysis seems to be robust and sufficiently explained.
Results
Generally, the results were presented in an organized and clear way and described well what is shown in the tables.

Discussion
This section requires improvement as it lacks sufficient discussion of the key findings. Instead, it primarily reiterates the results and focuses on how they align with previous studies. A more in-depth of the significance and implications of the findings is needed to strengthen the discussion.

Conclusion:
The authors in some places reiterate their results without adequately emphasizing their significance. Are there any practical implications of the study? Could the authors propose realistic recommendations for applying or managing these findings? This would enhance the impact and relevance of the work.

Experimental design

The research question has been carefully defined to address a specific knowledge gap in the field. It highlights the relevance and significance of understanding the association patterns of rhizosphere microbial communities with different Salix genotypes. The study aims to contribute meaningful insights into plant-microbe interactions and their spatial variability, filling a critical gap in current research.

The study was conducted with rigorous methodology and adhered to high technical and ethical standards. Care was taken to ensure the reliability of the experimental design, data collection, and analysis, meeting the scientific and ethical requirements of the field.

The methods are described in sufficient detail to ensure replicability. All procedures, materials, and analytical techniques were outlined clearly, providing the necessary information for other researchers to reproduce the study

Validity of the findings

While the impact and novelty of the study were not explicitly emphasized, the rationale and benefits of replicating this research are clearly stated. The study's findings provide valuable insights into plant-microbe interactions in the rhizosphere, and further replication could expand on these results to enhance their relevance and applicability in related fields.

All underlying data have been included in the study, ensuring robustness and statistical soundness. Appropriate controls and analytical methods were employed to validate the findings and enhance their reliability

Additional comments

No comment

---

## Round 0.2 · accepted · Accept

The study can be accepted now.

Reviewer 1 ·

Basic reporting

no comment

Experimental design

no comment

Validity of the findings

no comment

Additional comments

All comments were addressed, I suggested publication

Reviewer 3 ·

Basic reporting

The authors have thoroughly revised the manuscript in accordance with the feedback provided, addressing all the necessary changes and improvements. The revisions have been carefully implemented, and the manuscript is now ready for publishing.

Experimental design

The authors have carefully revised the Experimental design and response to the provided feedback, incorporating all necessary changes and improvements.

Validity of the findings

All underlying data have been provided and are robust, statistically valid, and well-organized.
The conclusions are clearly articulated, directly tied to the original research question, and focused solely on the supporting results